# Illuminating understudied kinases: a generalizable biosensor development method applied to protein kinase N
Julius Bogomolovas [ID] & Ju Chen [ID] [✉]

Protein kinases play crucial roles in regulating cellular processes, making real-time visualization of their activity essential for understanding signaling dynamics. While genetically encoded fluorescent biosensors have emerged as powerful tools for studying kinase activity, their development for many kinases remains challenging due to the lack of suitable substrate peptides. Here, we present a novel approach for identifying peptide substrates and demonstrate its effectiveness by developing a biosensor for Protein Kinase N (PKN) activity. Our method identified a new PKN substrate peptide that we optimized for use in a fluorescent biosensor design. The resulting biosensor shows specificity for PKN family kinases and can detect both overexpressed and endogenous PKN activity in live cells. Importantly, our biosensor revealed sustained basal PKN2 activity at the plasma membrane, identifying it as a PKN2 activity hotspot. This work not only provides a valuable tool for studying PKN signaling but also demonstrates a promising strategy for developing biosensors for other understudied kinases, potentially expanding our ability to monitor kinase activity across the human kinome.

Protein kinases found in all three domains of life and viruses regulate virtually every cellular process in living organisms[1–5]. Thus, real-time visualization and quantification of kinase activity in living cells is a cornerstone for both fundamental research and practical applications. As summarized in numerous reviews, genetically encoded fluorescent biosensors (GEFB) have emerged as the gold standard for studying kinase activity in live cells and organisms[6–11]. GEFBs offer a unique advantage over traditional biochemical assays. Unlike endpoint assays or in vitro kinase activity measurements, GEFBs allow for real-time, continuous monitoring of kinase activity in intact cellular environments with retained spatial and temporal resolution. At their core, kinase activity GEFBs consists of two components: a sensing unit detecting the signaling event and a reporting unit converting this change into quantifiable optically detectable readout. The sensing unit typically includes a phosphoresidue-binding domain and the peptide substrate phosphorylatable by kinase of interest[7,8]. The most common reporting unit readout in kinase activity GEFBs is based on in Förster Resonance Energy Transfer (FRET). FRET-based GEFBs are engineered by inserting sensing unit between two fluorescent proteins (FP), that form a FRET pair, which utilizes phosphorylation-induced conformational change to modulate energy transfer between FPs[8,11]. Significant improvements in biosensor performance have been achieved through optimization of the linkers connecting sensor units and selection and ordering of FPs with respect to other biosensor components[12–15].

A major breakthrough in kinase activity GEFB design came from development of the ExRai (excitation ratiometric) sensor[16]. This novel approach uses circularly permuted FP as the reporting unit, resulting in phosphorylation-dependent ratiometric changes of emission upon excitation at two different wavelengths. ExRai sensors have demonstrated unprecedented sensitivity compared to FRET-based activity sensors for kinases such as PKA, PKC, AMPK, or Akt[16–20].

Despite advancements in reporting module design, the development of new kinase biosensors faces a significant bottleneck: the identification of suitable peptide substrates. As illustrated in Fig. 1, out of the over 500 kinases[21] that make up the human kinome, we currently have sensors for just over 100. This underscores that while reporting and phosphopeptide-sensing modules have been optimized, the limitation now lies in finding appropriate peptide substrates for the vast majority of kinases. Efforts to address this challenge have involved deriving substrate peptides from various sources, including synthetic peptide libraries[22], published kinase substrates[12], and peptide consensus sequences from in vitro screening trials[23,24]. However, the identification of optimal peptide substrates remains a critical hurdle, hindering the development of sensors for many understudied kinases and leaving significant gaps in our ability to monitor kinase activity across the human kinome.

In this study, we present a novel, streamlined approach for identifying new peptide substrates for use in kinase biosensors. We demonstrate the

Department of Medicine, UCSD, La Jolla, CA, USA. [✉]e-mail: juchen@ucsd.edu

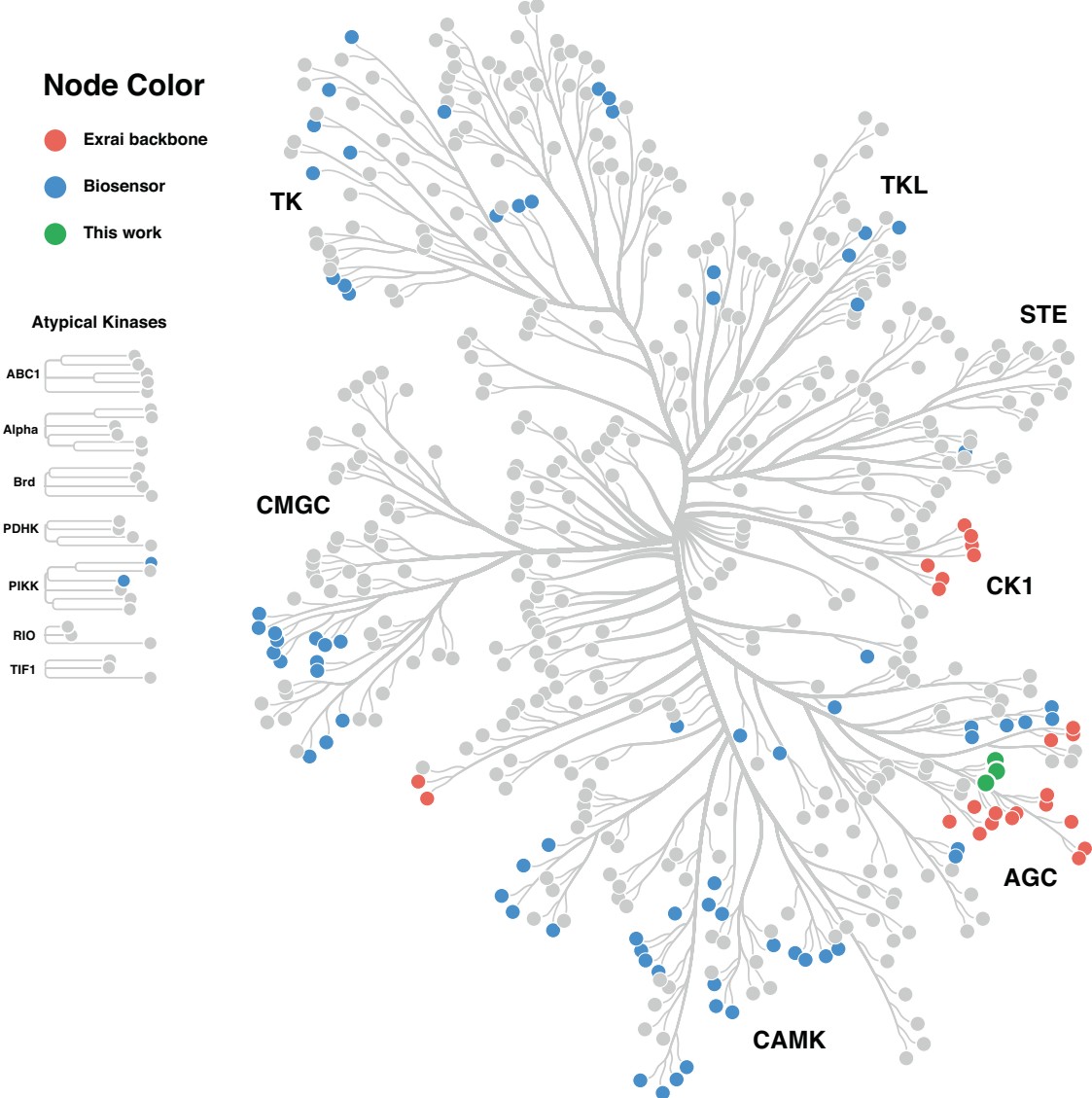

**Fig. 1 | Human kinome tree highlighting genetically encoded fluorescent biosensors (GEFBs) coverage.** This tree diagram illustrates the human kinome, with each node representing a distinct kinase. Color-coding indicates GEFB availability: gray nodes show kinases without sensors, blue nodes represent existing biosensors, red nodes denote kinases with ExRai backbone-based GEFBs, and green nodes highlight PKN kinases with newly developed biosensors from this study. Major kinase families are labeled (TK, TKL, STE, CMGC, CK1, AGC, CAMK).

effectiveness of this method by developing a biosensor for Protein Kinase N (PKN) activity. Scientific evidence summarized in numerous reviews underscores the broad spectrum of PKN kinase importance. PKNs play crucial roles in various cellular processes, including actin cytoskeleton reorganization, cell motility, cell proliferation, apoptosis, transcription regulation, and stress responses, and could act as potential targets for cancer, or neurodegenerative disorders[25–27]. While activation loop phosphorylation is widely used as a readout for PKN activation, accumulating evidence indicates that this is rather a marker of kinase maturation than activation[28]. Therefore, the need for a universal PKN activity marker is high.

In this work, we report a GEFB to monitor PKN activity in cells. This novel biosensor provides a powerful tool for real-time visualization of PKN dynamics, enabling deeper insights into its roles in various cellular processes. By addressing the challenge of substrate identification for an understudied kinase family, our work not only advances the understanding of PKN signaling but also demonstrates a promising strategy for developing biosensors for other kinases with limited known substrates. This advancement paves the way for more comprehensive studies of kinase activity across the human kinome, potentially uncovering new therapeutic targets and

improving our understanding of cellular signaling networks in both physiological and pathological states.

## Results

### Design of PKN activity sensor ExRai-PKNAR

To identify a suitable peptide for a GEFB targeting PKN activity, we employed a modified KESTREL approach (Fig. 2A). The KESTREL method involves saturating a pool of potential kinase substrates, such as tissue lysate, with the kinase of interest and ATP, followed by incubation and subsequent substrate identification[29]. We simplified the original protocol, which used chromatographic fractionation and manganese instead of magnesium in kinase reaction to supress background phosphorylation. Our modification involved heat inactivation of endogenous kinases by heating after dephosphorylation with λ phosphatase, a technique previously successful used in identifying Aurora kinase substrates[30]. Given that loss of regulatory domains activates PKNs[31,32], we utilized the isolated catalytic PKN2 kinase domain expressed in insect cells for this assay. Dephosphorylated and heat-inactivated lysate from the murine fibroblast cell line NIH3T3 served as the source of potential substrates. We used the wild-type (PKN2$^{WT}$) catalytic

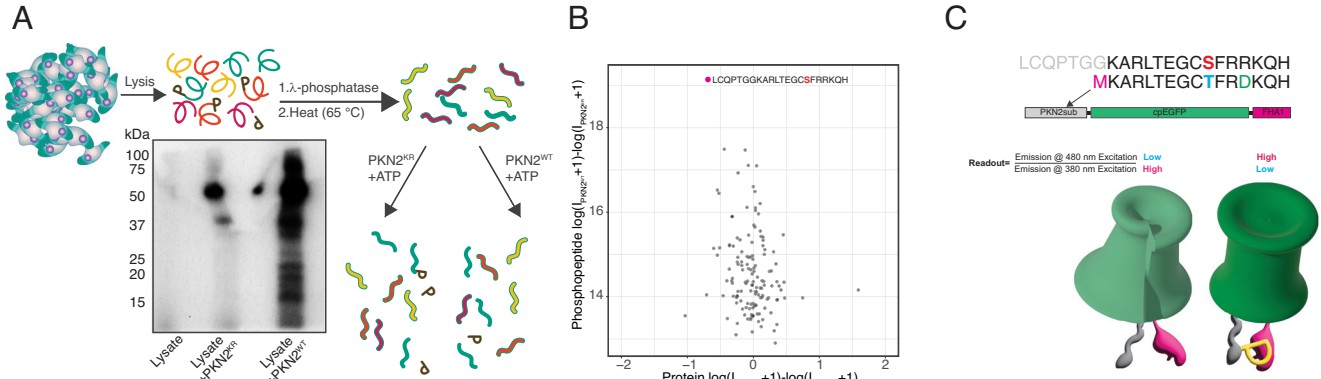

**Fig. 2 | Identification and optimization of a substrate peptide for PKN Activity biosensor. A** Modified KESTREL approach for substrate identification. NIH3T3 lysates were dephosphorylated, heat-inactivated, and incubated with PKN2 catalytic domain. Autoradiograph shows phosphorylation patterns. **B** Mass spectrometry analysis identifying enriched phosphopeptide (red) derived from ribosomal protein eS27. **C** Optimization of identified peptide for ExRai biosensor design, showing modifications and schematic of ExRai-PKNAR construct with expected fluorescence changes upon phosphorylation.

domain of PKN2 and a kinase-dead mutant of PKN2, where the catalytic lysine was mutated to arginine (PKN2$^{KR}$), as a negative control.

A small fraction of the reaction mixture was supplemented with γ-$^{32}$P-ATP, separated by SDS-PAGE, and exposed to a screen to detect phosphorylation. As anticipated, the inactivated lysate showed no phosphotransfer activity, the kinase-dead mutant exhibited only slight background, and the PKN2$^{WT}$ displayed strong phosphorylation (Fig. 2A). The remaining mixture, without radioactively labeled ATP, underwent mass spectrometry (MS) analysis aimed to quantify total and phosphoproteome composition. MS identified a phosphopeptide derived from Small ribosomal subunit protein eS27 as the most significantly enriched in the sample treated with PKN2$^{WT}$, while the abundance of mother protein was not different between compared samples (Fig. 2B).

Based on these results, we selected Small ribosomal subunit protein eS27 derived peptide as our candidate for GEFB development. To adapt it to the ExRai design, we shortened the peptide, replaced the phosphorylated serine with threonine, and introduced an aspartate at the +3 position to accommodate the contextual properties of the FHA1 domain used as the phosphoresidue binding element in ExRai. We then replaced the original PKA substrate peptide in ExRai-AKAR2[20] with this modified eS27-derived peptide to create a novel PKN activity reporter (PKNAR). Like ExRai-AKAR2, PKNAR is designed to provide a ratiometric readout based on the ratio of fluorescence emissions when excited at two different wavelengths - 480 nm and 380 nm. We anticipated that PKNAR would exhibit an increase in the 480/380 nm excitation ratio upon PKN activation and phosphorylation of its substrate peptide, similar to how ExRai-AKAR2 responds to PKA activation (Fig. 2C).

## Characterization of ExRai-PKNAR

To characterize ExRai-PKNAR, we conducted a series of experiments in HEK293T cells, using a Leica SP8 confocal microscope for all imaging. For each individual cell at each time point, the emission ratio when exciting at 488 and 405 nm was calculated as a readout (R = F488/F405). Following the standard practice in the field of biosensor development[16,20], these fluorescence intensity ratios (R) were then normalized to the baseline value before stimulation (R/R$_0$) for each cell to enable comparison across different experimental conditions, while accounting for cell-to-cell variability. Firstly, to validate that the peptide change altered the biosensor's specificity, we first compared the responses of ExRai-PKNAR and ExRai-AKAR2 in HEK293T cells stimulated with 50 μM forskolin/100 μM 3-isobutyl-1-methylxanthine (Fsk/IBMX) to induce PKA activity. As anticipated, ExRai-PKNAR showed no response to Fsk/IBMX stimulation, while ExRai-AKAR2 exhibited a strong response (Fig. 3A). Given the similarity between the catalytic domains of PKN and PKC family kinases[25], we also tested ExRai-PKNAR's reactivity to PKC activation. Stimulation with Phorbol 12,3-dibutyrate (PBDu), an established PKC activator[22], elicited no

response in ExRai-PKNAR expressing cells, indicating that the sensor does not react to PKC activity.

To assess ExRai-PKNAR's reactivity towards PKN activity, we co-transfected HEK293T cells with ExRai-PKNAR and mCherry-fused catalytic domains of all PKN family members, using mCherry alone as a negative control. We chose to use isolated catalytic domains as they represent maximally activated forms of PKN kinases. This approach is supported by previous studies showing that removal of N-terminal regulatory domains leads to dramatic increases in PKN activity, effectively mimicking the physiologically relevant activation state achieved through caspase-mediated cleavage of PKNs[31,32]. The 488/405 nm excitation ratios were significantly higher when ExRai-PKNAR was co-expressed with any PKN catalytic domain compared to the mCherry control, indicating that ExRai-PKNAR is responsive to activity of overexpressed PKNs (Fig. 3B). Notably, due to the high similarity and functional exchangeability of PKN family catalytic domains[25,33], ExRai-PKNAR reported activity of all PKN family members.

We next characterized ExRai-PKNAR's response to PKN2 activity in greater detail by examining the biosensor's temporal response using selective PKN2 chemical inhibitor[34]. Analysis of unnormalized excitation ratios (Fig. 3C) revealed that cells expressing ExRai-PKNAR and PKN2$^{WT}$ showed high baseline biosensor readout that decreased to the low levels seen with our negative controls upon inhibitor addition. Our negative controls - cells expressing either kinase-dead PKN2$^{KR}$ or the non-phosphorylatable ExRai-PKNAR$^{TA}$ - maintained consistently low ratios throughout the experiment. To account for cell-to-cell variability in sensor expression, we normalized the data to initial values for each cell (R/R$_0$), which showed a clear time-dependent reduction in normalized ratio upon inhibitor addition in cells expressing PKN2$^{WT}$ and ExRai-PKNAR, with no significant changes in our negative controls (Figs. 3D–F).

To further validate our biosensor's ability to reflect PKN2 activity, we performed immunoprecipitation- in vitro phosphorylation assay under matched experimental conditions - using the same cell line and identical wild-type and kinase-dead PKN2 constructs as in our biosensor experiments described above. In these parallel assays, immunoprecipitated wild-type PKN2 showed robust phosphorylation of substrates in dephosphorylated, heat-inactivated HEK293T cell lysate, while kinase-dead PKN2 exhibited no pronounced activity (Fig. 3G), paralleling biosensor readouts. In conclusion, these experiments demonstrate that ExRai-PKNAR is a specific and sensitive biosensor for PKN family kinases, capable of detecting changes in cellular activity of overexpressed PKN kinases.

## ExRai-PKNAR reveals sustained basal activity of PKN2 at plasma membrane

Development of ExRai-PKNAR opened new avenues to study spatiotemporal patterns of PKN activity with cellular resolution. PKN family

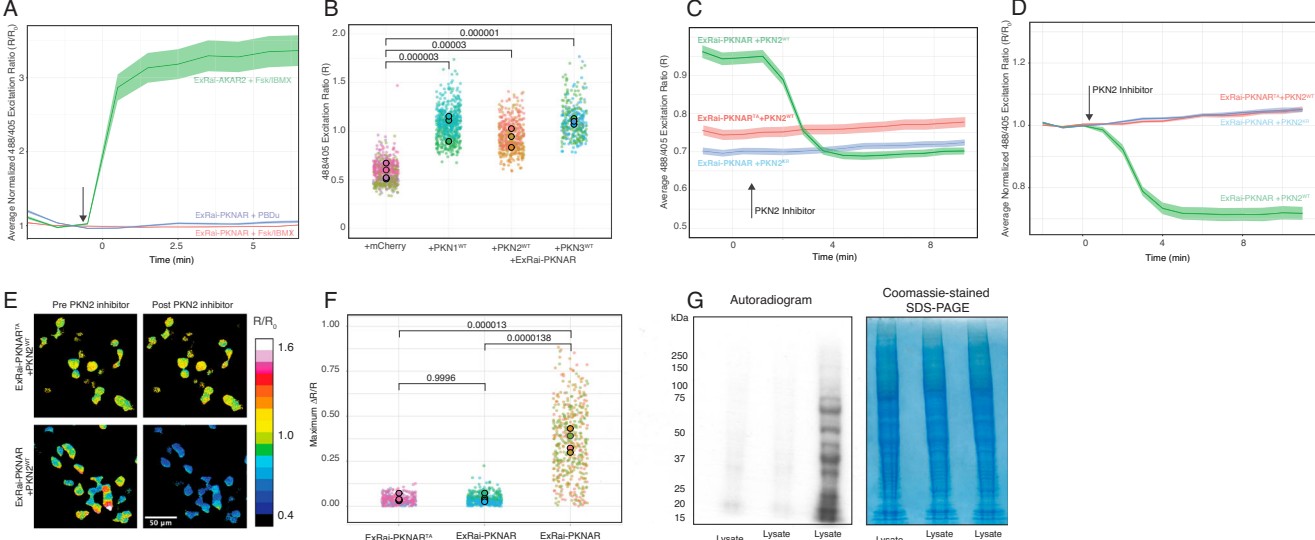

**Fig. 3 | Characterization of ExRai-PKNAR. A** Comparison of ExRai-PKNAR and ExRai-AKAR2 responses to PKA and PKC activation in HEK293T cells. The central line represents the mean values, with the shaded ribbon indicating the 95% confidence interval around the mean. **B** ExRai-PKNAR 488/405 nm excitation ratios when co-expressed with catalytic domains of PKN kinases or mCherry control. Each small dot represents a single cell measurement, with colors indicating different replicates. Larger dots show the mean for each replicate. Statistical analysis was performed using a generalized linear mixed model ($\chi^2(3) = 99.68$, $p < 0.001$), followed by post-hoc comparisons. All PKN catalytic domains caused significantly higher biosensor response compared to mCherry control (all $p < 0.0001$). Number of cells analyzed: mCherry ($n = 506$), PKN1 ($n = 508$), PKN2 ($n = 426$), PKN3 ($n = 308$), from 3 to 5 independent experiments. **C** Time course of non-normalized ExRai-PKNAR response to PKN2 inhibitor in cells expressing wild-type or kinase-dead PKN2. The central line represents the mean values, with the shaded ribbon indicating the 95% confidence interval around the mean. **D** Time course of normalized ExRai-PKNAR response to PKN2 inhibitor in cells expressing wild-type (PKN2$^{WT}$) or kinase-dead PKN2 (PKN2$^{KR}$). The central line represents the mean values, with the shaded ribbon indicating the 95% confidence interval

around the mean. **E** Representative pseudocolor images of ExRai-PKNAR before and after PKN2 inhibitor treatment. **F** Quantification of maximum ExRai-PKNAR responses for wild-type, kinase-dead, and phospho-site mutant versions. $\Delta R$ was calculated as $abs(R_{max} - R_{min})/R_{min}$, where $R_{max}$ and $R_{min}$ are the average fluorescence ratios (F488/F405) before and after PKN2 inhibitor addition, respectively. Each dot represents a single cell measurement, with colors indicating different replicates. Larger dots show the mean for each replicate. Statistical analysis was performed using a generalized linear mixed model ($\chi^2(2) = 90.89$, $p < 0.001$), followed by pairwise comparisons with Tukey's adjustment. Only the wild-type sensor + wild-type kinase combination showed significant response to inhibitor treatment ($p < 0.0001$). Number of cells analyzed: ExRai-PKNAR + PKN2$^{WT}$ ($n = 452$), ExRai-PKNAR + PKN2$^{KR}$ ($n = 390$), ExRai-PKNAR$^{TA}$ + PKN2$^{WT}$ ($n = 303$), from 4 independent experiments. **G** Validation of PKN2 activity using immunoprecipitation-in vitro phosphorylation assay. PKN2$^{WT}$ and kinase-dead PKN2$^{KR}$ were immunoprecipitated from HEK293T cells and incubated with dephosphorylated, heat-inactivated cell lysate in the presence of $\gamma$-$^{32}$P-ATP revealing robust phosphorylation activity only in the PKN2$^{WT}$ sample, confirming correlation with biosensor readouts.

members contain regulatory domains, which can position PKNs at plasma through various interactions[25] and is functionally relevant[35,36]. Immunofluorescence staining revealed that endogenous PKN2 is both diffusely distributed in the cytoplasm and localized to the plasma membrane in HEK293T cells, suggesting the existence of distinct cellular pools (Fig. 4A). To investigate whether PKN2 exhibits localization-dependent activity patterns, we generated plasma membrane targeted ExRai-PKNAR (PM-ExRai-PKNAR) using N-terminal lipid modification motif derived from Lyn kinase and confirmed its proper localization in HEK293T cells (Fig. 4A lower panel). As a negative control we mutated phosphorylatable threonine to alanine in same biosensor setup (PM-ExRai-PKNAR$^{TA}$). HEK293T cells were transfected and imaged using spinning disc confocal microscopy. Inhibition of endogenous PKN2 activity upon addition of 10 μM PKN2 inhibitor did not elicit significant changes in cells expressing non-targeted (ExRai-PKNAR) or inactive version of membrane targeted (PM-ExRai-PKNAR$^{TA}$) biosensor. However, inhibition of PKN2 activity caused significant drop in readout signal for membrane targeted PKN biosensor (PM-ExRai-PKNAR) (Fig. 4B–D). The lack of significant changes in cells expressing the non-targeted ExRai-PKNAR upon PKN2 inhibition suggests that the overall cytosolic PKN2 activity is relatively low under basal conditions. This contrasts sharply with the sustained activity observed at the plasma membrane, identifying plasma membrane as a PKN2 activity hotspot. The differential response between cytosolic and membrane-targeted biosensors to PKN2 inhibition corroborates the immunofluorescence data, confirming the existence of distinct pools of PKN2, potentially allowing for compartment-specific regulation of PKN2-mediated signaling pathways.

## Assessment of different substrate identification strategies for PKN2 biosensor

To validate our substrate identification strategy, we compared our KESTREL-derived peptide against two alternative approaches for substrate selection. We evaluated our PKNAR peptide, against computationally predicted optimal PKN2 substrate based on recent Ser/Thr kinase specificity atlas[37], and a previously biochemically validated PKN substrate peptide derived from CLIP-170 protein[38,39]. We assessed their performance in HEK293T cells co-expressing mCherry-PKN2$^{WT}$ by monitoring biosensor response to PKN2 inhibitor treatment. The computationally predicted sequence scored in the highest percentile (>99th) for predicted substrate efficiency among 82,735 phosphorylation sites, while our KESTREL-derived peptide showed somewhat lower prediction scores, and the CLIP-170 peptide ranked worst in computational predictions (Fig. 5B, C). However, actual performance in living cells showed a different pattern. Upon PKN2 inhibitor treatment, our KESTREL-derived peptide demonstrated the largest and most consistent change in biosensor signal, the computationally predicted optimal substrate showed moderate response, and the CLIP-170 peptide exhibited virtually no change in the biosensor readout (Fig. 5A). These results empirically validate our modified KESTREL approach and highlight that neither computational predictions nor prior biochemical validation in cell-free assays reliably predict a peptide's performance in the context of a conformational biosensor in living cells.

## Discussion

Despite the enormous scientific potential of GEFBs for kinase activity, most human kinome members still lack dedicated sensors (Fig. 1). Our analysis

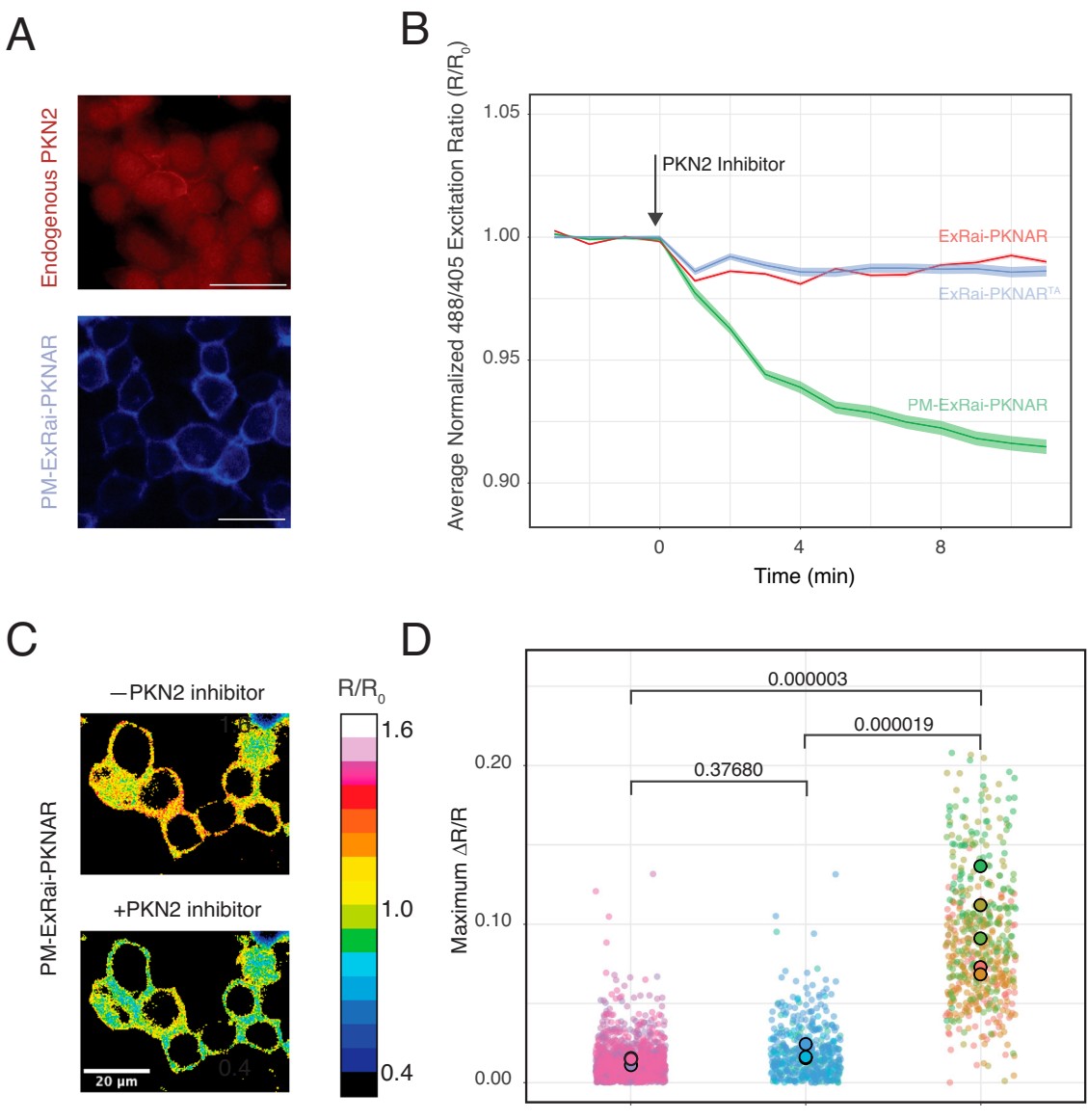

**Fig. 4 | Localization-dependent PKN2 activity in HEK293T cells. A** Localization of endogenous PKN2 (upper panel) and membrane targeted ExRai-PKNAR emission upon 405 nm excitation in HEK293T cells. **B** Time course of ExRai-PKNAR responses to PKN2 inhibitor for cytosolic, membrane-targeted, and control biosensors. Inset shows immunofluorescence of endogenous PKN2 localization. The central line represents the estimated values, and the surrounding shaded ribbon indicates the 95% confidence interval. **C** Representative pseudocolor images of plasma membrane-targeted ExRai-PKNAR before and after PKN2 inhibitor treatment. **D** Quantification of maximum biosensor responses for cytosolic, membrane-targeted, and control ExRai-PKNAR variants. Each dot represents a single cell measurement, with colors indicating different replicates. Larger dots show replicate means. Statistical analysis was performed using generalized linear mixed model ($\chi^2(2) = 130.41$, $p < 0.001$) with Tukey's post-hoc comparisons. The membrane-targeted biosensor showed significantly higher response compared to both cytosolic ($p < 0.0001$) and inactive ($p < 0.0001$) variants, while cytosolic and inactive variants did not differ significantly ($p = 0.3549$). This reveals sustained PKN2 activity at the plasma membrane that can be inhibited. Cell numbers: LRay ($n = 549$), ray ($n = 1247$), RayTA ($n = 445$), from 3 to 5 independent experiments.

reveals that while approximately 100 kinases have associated biosensors, the number of unique peptide substrates used in biosensors is considerably lower due to cross-reactivity between kinase isoforms. This shortage underscores the need for new substrate peptides to expand biosensor coverage and improve specificity, a challenge our approach is well-positioned to address.

The method we employed for identifying a suitable peptide substrate, while seemingly crude in its destruction of cellular architecture, proved highly effective. This approach offers a key advantage: it denatures proteins, removing their 3D context, which aligns well with our goal of using a peptide rather than a whole protein in biosensor. However, the heat treatment also precipitates some proteins, potentially limiting kinase access to specific substrates. A potential refinement could involve dephosphorylation followed by trypsin digestion and subsequent rephosphorylation, which might expose a broader range of peptide substrates while maintaining the benefits of denaturation. Despite this limitation, our approach's success suggests it could be readily applied to develop biosensors for other understudied kinases.

To validate our substrate identification approach, we compared the performance of the ExRai-PKNAR biosensor by replacing the substrate peptide with a candidate selected from the Ser/Thr kinase substrate specificity atlas[37] and the CLIP-170 peptide, which has been previously validated as a PKN model substrate[38,39]. Despite the CLIP-170 peptide showing robust phosphorylation in solution-based assays, it produced minimal response in

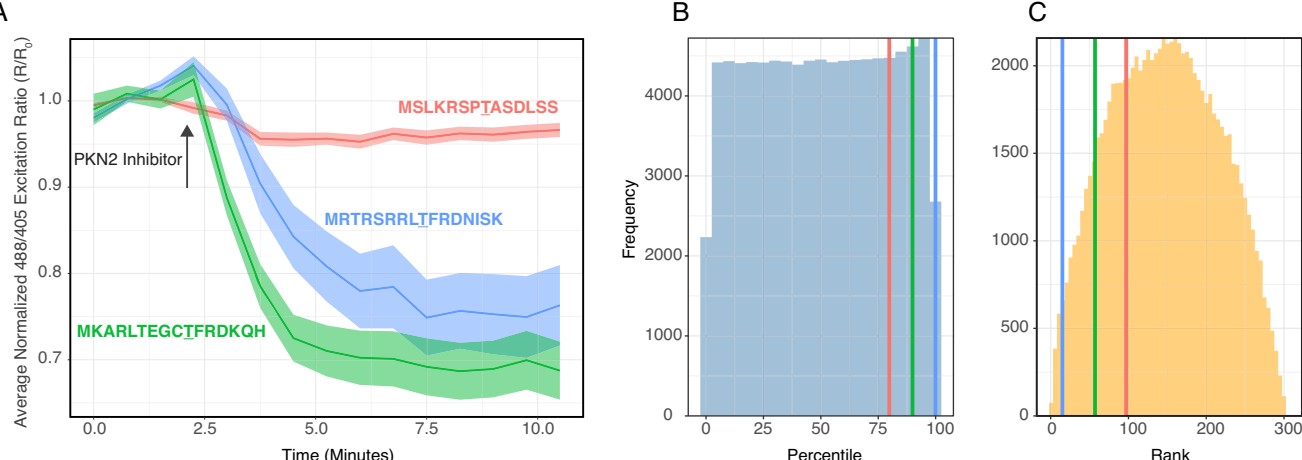

**Fig. 5 | Comparison of substrate identification approaches for PKN2 biosensor development. A** Time course of biosensor responses to PKN2 inhibitor (10 μM) in HEK293T cells co-expressing mCherry-PKN2^WT and biosensor variants. Three different substrate peptides were tested, with their sequences shown in matching colors: KESTREL-derived PKNAR peptide (green), computationally predicted optimal sequence (blue), and literature-derived PKN substrate (red). Lines represent mean response, with shaded areas showing 95% confidence interval. Cell numbers: literature-derived PKN substrate ($n = 95$), computationally predicted ($n = 57$), KESTREL-derived ($n = 85$). **B** Percentile distribution showing where each peptide ranks within PKN2's scoring distribution across 82,735 phosphorylation sites. Vertical lines in matching colors indicate the percentile score for each peptide sequence. **C** Comparative ranking of each peptide's specificity score against 302 other kinases, demonstrating relative preference for PKN2 versus other kinases. Lower rank numbers indicate higher specificity for PKN2. Vertical lines in matching colors indicate the rank of each peptide, with distribution showing overall ranking patterns in the dataset.

the biosensor context. The peptide predicted using Ser/Thr kinase substrate specificity atlas as the preferred PKN2 substrate performed moderately well; however, our peptide produced the strongest biosensor response despite having lower predicted substrate efficiency. For researchers considering whether a known kinase substrate peptide might work in a biosensor context, the Ser/Thr kinase substrate specificity atlas-based substrate efficiency prediction tool can guide peptide selection. However, our phosphoproteomics approach identified a peptide that outperformed the theoretically optimal substrate in actual biosensor applications despite slightly lower computational predictions. This observation suggests that while computational predictions and traditional biochemical characterization can guide peptide selection for kinase activity biosensors, our tailored phosphoproteomics-based approach resulted in the best peptide substrate for biosensor.

Our experiments with both overexpressed and endogenous PKN2 reveal important insights about PKN2 regulation. When overexpressing just the catalytic domain of PKN2, which lacks regulatory domains, we observe kinase activity using an untargeted, cytoplasmic version of ExRai-PKNAR. This is expected, as removal of regulatory domains is known to activate PKNs[31,32], and lacking the domains that mediate membrane binding, the overexpressed catalytic kinase domain distributes throughout the cytoplasm[25]. However, the behavior of endogenous PKN2 tells a different story. Here, the full-length kinase, with all its regulatory domains intact, shows a compartmentalization of activity. Despite being present in both cytoplasmic and membrane pools, only the membrane-associated PKN2 pool elicits an ExRai-PKNAR response upon inhibition. The differential activity we detected between membrane-associated and cytoplasmic PKN2 pools aligns with previous findings on PKN1. Earlier study using biochemical approaches suggested that the plasma membrane pool of PKN1 is more active than its cytosolic counterpart[36].

Our inhibition-based biosensor development and validation approach highlights important methodological and conceptual considerations in kinase activity studies. The prevalent experimental paradigm in kinase activity studies, especially using GEFBs, follows an activation sequence: cells are first frequently serum-starved to establish a minimal baseline activity, then stimulated with specific activators, and optionally treated with inhibitors to confirm signal specificity. This approach has proven valuable for understanding the spatiotemporal dynamics of kinases such as PKA, PKC,

and Akt[12,13,16,17,19,20]. In contrast, we employed an inhibition-based approach, starting with unperturbed cells and applying a specific PKN2 inhibitor. We initially used this strategy for initial biosensor characterization, as it offered a straightforward validation using constitutively active PKN catalytic domains, similar to approaches used in improving Fyn and ZAP70 kinase biosensors[40]. When we extended this inhibition paradigm to study endogenous PKN2, we discovered substantial constitutive activity, specifically at the plasma membrane. Similar patterns of constitutive activity have been observed in studies of ROCK kinase using GEFBs, where inhibitor treatment causes biosensor signals to decrease substantially below baseline levels[23,41]. While both PKN2 and ROCK kinases can associate with the plasma membrane, their activity patterns show a structural distinction. ROCK kinases possess an extended coiled-coil domain (~100 nm) that separates their catalytic domain from membrane-binding regions, allowing their activity to be detected by cytoplasmic biosensors even when membrane-bound. In contrast, PKNs lack such a spacer domain, making their catalytic activity more strictly confined to the membrane compartment. Our findings suggest that when developing new kinase biosensors, both activation and inhibition paradigms should be considered, as constitutive kinase activity - potentially compartmentalized by protein architecture - may be overlooked by traditional activation-based approaches.

The pattern of constitutive PKN2 activity may provide insight into why it is essential for mouse embryonic viability. Among PKN isoforms, only PKN2 deletion results in embryonic lethality[42], mirroring another constitutively active kinase, ROCK2[43,44]. This stands in contrast to kinases that require specific activation signals resulting in dynamic activity patterns, such as PKC family members, where only simultaneous knockout of several isoforms leads to embryonic lethality[45]. This pattern suggests that constitutively active kinases may be less functionally redundant, making their loss more detrimental to organism homeostasis.

What is the functional significance of PKN2's membrane-localized activity? Previous studies have shown that PKN2 depletion impairs migration in both cancer and normal cells through its role in organizing actin filaments at the cell rear[33,42,46,47]. While our current studies in HEK293T cells, which lack clear polarization and have minimal cytoskeletal organization[48], revealed constitutive membrane activity, an important next step will be examining whether PKN2 activity shows spatial patterns along the plasma membrane in polarized, migrating cells. Additionally, PKN2

plays a crucial role in forming cell-cell contacts, particularly apical junctions[35]. This dual role is particularly evident in cardiac development, where cardiomyocyte-specific PKN2 deletion results in severe malformation of the myocardium due to impaired myocardium compaction[49] - a process requiring precise coordination of both cell migration and intercellular connection formation[50,51]. Our biosensor provides a valuable tool for directly visualizing how PKN2 activity orchestrates these complex cellular behaviors.

While ExRai-PKNAR has already demonstrated high sensitivity and specificity, there is potential for further optimization. As demonstrated with the ExRai-AKAR2 sensor for PKA activity and ExRai-AktAR2 for Akt activity, screening of linker variants connecting cpGFP to sensing module can significantly enhance biosensor performance[19,20]. A similar approach could be applied to ExRai-PKNAR, potentially yielding even greater dynamic range and sensitivity.

In conclusion, our development of ExRai-PKNAR provides a valuable tool for studying PKN signaling and demonstrates a practical approach for expanding the repertoire of kinase activity biosensors. As we continue to fill in the gaps in our ability to visualize kinase activity across the human kinome, tools like ExRai-PKNAR will be crucial in unraveling the complex signaling networks underlying cellular processes in health and disease.

## Methods

### Survey of GEFBs for kinase activity
A complete list of human kinome members was obtained from KinMap[52]. For each kinase, a systematic literature search was conducted using PubMed and Google Scholar. Search terms included the kinase name combined with "fluorescent activity sensor" or "fluorescent activity biosensor". Reports were included if they reported a genetically encoded fluorescent biosensor capable of real-time kinase activity monitoring in living cells. Peer-reviewed publications, preprints and conference abstracts were considered. For each identified biosensor, the following information was evaluated: target, sensor backbone type, substrate peptide and isoform specificity. When isoform specificity was not explicitly stated, the sensor was assumed to detect all isoforms of the kinase family (e.g., AKT1, AKT2, and AKT3 for an Akt sensor). The search was concluded once at least one GEFB was identified for a given kinase. For kinases where no biosensors were identified, this was noted in the supplementary data. In cases where multiple biosensors were found for a single kinase, the first reported biosensor was typically selected for inclusion. However, if an ExRai-based sensor was available, it was prioritized over other designs regardless of publication order. The collected data were visualized using Coral[53]. Kinases with at least one reported GEFB were highlighted, distinguishing between ExRai-based and other sensor designs. Kinases without any identified biosensors were left unmarked in the visualization. The raw data collected during this analysis, including kinase names, associated biosensors, and relevant publications, are provided in Supplementary Data. This analysis was limited to publicly available information up to June 2024. While efforts were made to be comprehensive, it is possible that some biosensors, particularly those recently developed or not widely published, may have been overlooked.

### In vitro phosphorylation reactions
NIH3T3 murine fibroblasts (ATCC CRL 1658), grown in DMEM supplemented with 10% FCS and Normocin, were used as a source of potential PKN2 substrates. Cells were harvested and lysed in buffer containing 25 mM Tris-HCl pH 7.4, 150 mM NaCl, 5 mM MgCl2, 1 mM EDTA, 1% NP-40, and 5% glycerol, supplemented with Turbonuclease and protease inhibitors. Lysates were then supplemented with 2 mM MnCl$_2$ and 2 mM DTT, treated with lambda phosphatase at 30 °C for 30 min, and heat-inactivated at 65 °C for 10 min.

Recombinant wild-type and kinase-dead PKN2 catalytic domains (residues 645-983 referenced to NP_848769.2) were produced and purified from insect cells as described earlier[54]. In vitro phosphorylation reactions were set up by adding 1 mM ATP and the recombinant PKN2 catalytic domains to the heat-inactivated lysates.

A small portion of the reaction mixture was supplemented with γ-$^{32}$P-ATP for radioactive labeling. Reactions were incubated at 37 °C for one hour and terminated by adding EDTA to a final concentration of 13 mM. The non-radioactive reactions were subjected to mass spectrometry analysis, while the radioactive samples underwent SDS-PAGE, followed by gel drying and autoradiography.

For immunoprecipitation-in vitro phosphorylation assays, HEK293T cells expressing mCherry-PKN2 constructs were lysed and immunoprecipitated using Anti-FLAG® M2 Magnetic Beads (20 µL per sample) in immunoprecipitation buffer (25 mM Tris pH 7.4, 150 mM NaCl, 1% NP-40, 1 mM EDTA, 5% glycerol) supplemented with phosphatase and protease inhibitors. After 1-hour incubation at room temperature with gentle rotation, beads were washed three times with TBS (25 mM Tris pH 7.4, 150 mM NaCl). Heat-inactivated, dephosphorylated cell HEK293T lysate was prepared following the protocol described above and added to the immunoprecipitated kinases along with cold ATP and γ-$^{32}$P-ATP. The kinase reaction proceeded for 30 minutes at room temperature before being terminated with SDS loading buffer. Samples were separated by SDS-PAGE, stained with Coomassie Blue, dried, and exposed to autoradiography film.

### Phosphoproteomic analysis
Heat-inactivated in vitro phosphorylation reaction mixtures at a 2.6 mg/mL protein concentration were resolubilized in 8 M urea and 50 mM ammonium bicarbonate with Benzonase. Proteins were reduced with 5 mM TCEP, alkylated with 15 mM iodoacetamide, and digested overnight with Trypsin/Lys-C mix. Peptides were desalted using AssayMap C18 cartridges, and a fraction was enriched for phosphopeptides using Fe(III)-NTA cartridges.

LC-MS/MS analysis was performed using a Proxeon EASY-nanoLC system coupled to an Orbitrap Fusion Lumos Tribid mass spectrometer. Peptides were separated on an analytical C18 Aurora column using a 75-min gradient. The mass spectrometer was operated in positive data-dependent acquisition mode with 1-second cycles for survey and MS/MS scans. Mass spectra were analyzed using MaxQuant software (version 1.6.11.0) against the Mus musculus Uniprot protein sequence database and GPM cRAP sequences.

Carbamidomethylation of cysteine was set as a fixed modification, while oxidation of methionine, acetylation of protein N-terminal, and phosphorylation of S/T/Y were variable modifications. The false discovery rate for spectrum and protein identification was set to 1%. Due to the absence of replicates, a stringent analysis approach was employed. Only phosphopeptides with evidence at the protein level were considered. Furthermore, phosphopeptides were included in the analysis only if their corresponding unmodified proteins were detected at comparable levels between samples.

### Construction of ExRai-PKNAR and PKN expression plasmids
ExRai-PKNAR was generated by replacing the original PKA substrate in ExRai-AKAR2[20] (a generous gift from Dr. Jin Zhang) with the identified PKN substrate peptide. This was achieved using overlap extension PCR with PrimeSTAR GXL polymerase (Takara). The resulting PCR fragment was linearized and assembled using NEBuilder HiFi DNA Assembly (NEB). The construct was then transformed into NEB 5-alpha competent E. coli cells, amplified, and the plasmid was isolated. The sequence was verified by whole plasmid sequencing (Plasmidosaurus).

A non-phosphorylatable mutant (ExRai-PKNAR$^{TA}$) was created by mutating the phosphorylatable threonine to alanine. Additionally, a plasma membrane-targeted version was generated by introducing the N-terminal 11 amino acids from Lyn kinase (MGCIKSKRKDK) to the N-terminus of ExRai-PKNAR. These variants were constructed using the same overlap extension PCR and assembly approach, with ExRai-PKNAR as the template.

Catalytic domains of PKN family members and their inactive mutants were expressed as N-terminally FLAG tagged fusion proteins with mCherry using the pmCherry-C1 plasmid.

## Live-cell imaging and immunofluorescence

HEK293T (ATCC CRL-3216) cells were cultured in DMEM supplemented with 10% FCS and Normocin. For imaging experiments, cells were seeded onto polylysine-coated Ibidi µ-Slide 8 Well chambered coverslips. Cells were transfected 12 h after seeding using Lipofectamine 3000 according to the manufacturer's protocol. Imaging was performed 12–24 h post-transfection, after replacing the media with Live Cell Imaging Solution (Gibco, #A59688DJ).

For a subset of experiments, immunofluorescence was performed to visualize endogenous PKN2. Cells were fixed with 4% paraformaldehyde in PBS for 15 min, permeabilized with 0.1% Triton X-100 for 5 min and blocked with 3% BSA for 1 h. Cells were then incubated with anti-PKN2 primary antibody (1:200 dilution, Sigma-Aldrich HPA034861) overnight at 4 °C, followed by Alexa Fluor 647-conjugated secondary antibody (1:500 dilution, AffiniPure™ Donkey Anti-Rabbit IgG (H + L) Jackson ImmunoResearch Laboratories #711-605-152) for 1 h at room temperature.

Confocal imaging was conducted on a LEICA SP8 microscope using a 40X (1.10 NA) water immersion objective. Excitation wavelengths were 488 nm and 405 nm, with emission collected at 500-575 nm.

Spinning disc confocal imaging was performed on a Nikon Eclipse Ti2-E microscope equipped with a 20X (0.75 NA) water immersion objective. Excitation was achieved using 405 nm and 477 nm lasers, with emission collected using a 520/40 nm filter. Images were acquired at room temperature with a frame rate of 45–60 s.

## Image analysis

Acquired image stacks were processed using Fiji[55] software. To correct for lateral drift primarily caused by inhibitor addition, linear stack alignment with SIFT[56] was performed. The aligned stacks were then split into separate channels and converted to 32-bit images to allow for NaN background values. Background correction was implemented by applying thresholding to isolate pixels only from cells expressing the biosensor, with all non-expressing regions set to NaN values. No oversaturated pixels were observed in the analyzed datasets. The thresholded images were used to generate ratiometric images by dividing the longer excitation (477 or 480 nm) image by the shorter excitation (405 nm) image. Ratiometric images were generated by dividing the longer excitation (477 or 480 nm) image by the shorter excitation (405 nm) image.

Cell segmentation was performed using Cellpose[57]. For non-targeted ExRai-PKNAR, the *cyto3* model was applied to single-channel images using the longer excitation wavelength due to its stronger signal. For membrane-targeted biosensors, a fine-tuned model was developed using a human-in-the-loop training approach from *cyto2* model, specifically selecting cells with proper membrane localization of the biosensor.

Quantification of biosensor response was carried out using TrackMate[58] on the ratiometric images and generated cell masks. The mean ratio per cell per frame was calculated and exported as CSV files for further analysis. In some experiments, frame 0 was excluded from the analysis to account for initial stabilization of the signal.

## Statistics and reproducibility

Data analysis was performed using R (version 4.4.1) with the glmmTMB[59] package (version 1.1.9) for generalized linear mixed models (GLMMs). Raw excitation ratio values and maximum ratio changes were analyzed. Maximum ratio changes were calculated as $abs(R_{max} - R_{min})/R_{min}$, where $R_{max}$ and $R_{min}$ are the average values pre- and post-inhibitor addition, respectively. The absolute value was used to handle occasional small negative values in non-responsive conditions. GLMMs with a Gamma distribution and log link function were fitted to account for the nested structure of the data, with cells measured within experiments. We chose to use a Gamma distribution in our GLMMs as it provided a good fit for our non-negative, right-skewed data and is commonly used for modeling ratio data. Fixed effects included experimental conditions, while random effects accounted for the nested structure. In some cases, a dispersion formula for each nested variable was included to improve model fit. Model diagnostics were performed using the DHARMa[60] package (version 0.4.6) to assess residual normality and dispersion. While some deviations from normality were observed in the residuals, we proceeded with the analysis as GLMMs are known to be robust to minor violations of distributional assumptions[61]. To validate this choice, we performed sensitivity analyses using GLMMs with Gaussian and inverse Gaussian distributions on the non-absolute transformed data $(R_{max} - R_{min})/R_{min}$, which can accommodate negative values. These analyses yielded consistent statistical conclusions, though the Gamma distribution with absolute-transformed data provided superior model diagnostics. Post-hoc comparisons were conducted using the emmeans[62] package (version 1.10.3) with degrees of freedom conservatively estimated as the number of replicates minus one. Type III ANOVA was performed using the car[63] package (version 3.1-2). Statistical significance was set at $p < 0.05$ for all analyses. Data visualization was performed using the ggplot2[64] package (version 3.5.1). Quantitative biosensor activity data were presented in format of SuperPlots[65,66]. Sample sizes and definition of replicates provided in figure legends.

## Reporting summary

Further information on research design is available in the Nature Portfolio Reporting Summary linked to this article.

## Data availability

The datasets generated and/or analyzed during the current study are available from the corresponding author upon reasonable request. Numerical data used to generate figures can be accessed via Figshare: https://figshare.com/projects/Numerical_data_for_figures_in_Illuminating_Understudied_Kinases_A_Generalizable_Biosensor_Development_Method_Applied_to_Protein_Kinase_N/233066. Uncropped protein gels and corresponding autoradiograms are provided in supplementary materials. The plasmids described in this study have been deposited to Addgene: ExRai-PKNAR (#226631), ExRai-PKNAR(T/A) (#226632), PM-ExRai-PKNAR (#226633), and PM-ExRai-PKNAR(T/A) (#226634).

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

## Acknowledgements
This work was supported by NIH grant R01HL155826 to J.C. We thank Dr. Svetlana Maurya and The Sanford Burnham Prebys Proteomics Shared Resource for phosphoproteomic analysis. Microscopy was performed at the UCSD Microscopy Core (NINDS P30NS047101), with assistance from Jennifer Santini, and the Nikon Imaging Center at UC San Diego, with support from Dr. Peng Guo. We are grateful to Dr. Sohum Mehta and Dr. Jin Zhang for providing the ExRai-AKAR2 construct and their mentorship. We thank Iris Zaretzki for technical assistance.

## Author contributions
J.B. conceived the project, designed and performed experiments, analyzed data, developed analysis tools, prepared figures, and wrote the manuscript. J.C. secured funding and supervised the project. Both authors reviewed and approved the final manuscript.

## Competing interests
The authors declare no competing interests.
