## [Transparent Peer Review file · Communications Biology]

illuminating Understudied Kinases: A Generalizable Biosensor Development Method Applied to Protein Kinase N

Corresponding Author: Professor Ju Chen

This manuscript has been previously reviewed at another journal. This document only contains information relating to versions considered at Communications Biology.

Version 0:

Reviewer comments:

Reviewer #1

(Remarks to the Author)

In this study, Bogomolovas and Chen develop a new genetically encoded fluorescent biosensor for protein kinase N (PKN) using an optimized substrate identification method. This method probes the entire cell contents for potential kinase substrates using phosphoproteomics to overcome the bottleneck of substrate identification. The specificity of this biosensor is demonstrated by assessing the biosensor response when AGC family kinases are activated. The PKN biosensor shows increased responses when PKN is overexpressed and shows decreased response when inhibited. The biosensor is then used to examine differences in plasma membrane and cytosolic PKN activity in HEK293 cells, revealing greater activity at the membrane.

I commend the authors on a very nice piece of work. The substrate mining method is clearly effective, given that the major hit from the screen yielded a functional and specific biosensor when the substrate was ported into the established ExRai backbone. However, the paper must be further strengthened by additional experiments, particularly stimulus experiments to activate PKN and demonstrate the full dynamic range of the biosensor and comparisons to phosphorylation, which is still a gold standard for activation. Some of the figures should also be refined to include more raw values, as the data presentation makes it difficult to fully assess the performance of the biosensor against negative controls.

Major points:

1. PKN Activation. For any newly developed biosensor, a thorough characterization of its dynamic range between fully inhibited and fully activated contexts is essential. The authors use inhibitor experiments to examine the "off state" of the biosensor, but only use PKN overexpression, not activation, to stimulate this kinase. However, protein kinases are typically highly regulated, so overexpression alone is unlikely to yield a state of maximum PKN activity. An ideal experiment would stimulate PKN activity to show the full dynamic range of the biosensor, for example by introducing a constitutively active RhoA. The response to such a PKN activator could also serve as a positive control for Figure 3A.

In addition, including a ground truth measure of PKN2 activity such as a Western blot or immunostaining for phospho-PKN2 in parallel to stimulation/overexpression experiments would more comprehensively show how the biosensor's dynamic range compares to this classical measurement of activity.

2. Non-normalized visualizations of biosensor activity. The authors show normalized data in time course experiments in which inhibitors are applied (Figure 3C, Figure 4A). This clearly illustrates that biosensor output falls with reduced PKN2 activity. However, it does not speak to the dynamic range of the biosensor. The R/R_0 seems to be a normalized measurement to the value in the start frame of the time course, but not the baseline value (which R_0 usually stands for). It is a bit unclear for readers if $R/R_0 = 1.00$ is a state where PKN is fully activated or if it is baseline with low PKN activity. How does the baseline output of the wild-type PKN2 + biosensor compare to the non-catalytic PKN2KR case, or the non-phosphorylatable ExRai-PKNARTA biosensor? For example, in figure 3C, it is nice and clear to see the control 'TA+PKN-WT' show no change upon drug inhibition, but it does not show if TA-mutant biosensor has a lower baseline output than a WT biosensor due to the normalization. To address that, it would be nice to show an unnormalized version of the plots, where WT-biosensor starts high and then goes down after drug inhibition,

while the controls start low and keep low all along. It could be also useful to show the F/F0 measurements separately in 488 and 405 channels upon drug inhibition.

3. Motivation and discussion of PKN2. The authors describe the myriad roles that PKN plays in cellular function, however, this description did not feel compelling in motivating their choice to pursue creation of a PKN biosensor or their investigations into spatial differences in activity. It would greatly strengthen this paper if the authors further elaborated on the significance of PKN2 and their findings. Is there a canonical cell behavior that PKN2 is known to be involved in that motivates this work? Given the findings that PKN2 is active at the plasma membrane, is there evidence regarding what role it may be playing there? Is there a biological context in which PKN2 activity at the membrane is relevant?

4. Evidence on correct biosensor subcellular localization. In Figure 4B, normalized images are shown of the plasma-membrane-targeted biosensor. The findings regarding PKN activity differences in different subcellular locations should be substantiated by including raw images in either 488 or 405 channels to show that the PM-biosensor is actually localized on the membrane as a supplement to Figure 4B.

Minor points

1. Potential mislabelling in Figure 4A. The authors mention in text that one control is “inactive version of membrane targeted (PM-ExRai-PKNAR-TA)” (line 137-139). However the time course curve in Figure 4A might be mislabeled as “ExRai-PKNAR-TA”. If this is not mislabeled, an PM-ExRai-PKNAR-TA is an appropriate negative control to include in this figure.

2. Demonstrate if the substrate consensus exists for PKN. There seems to be many hits showing strong phosphopeptide enrichment in WT-PKN pool from Figure 2B. The authors pick the top one for downstream biosensor development. I'm curious, for the top substrate hits that came out of your mass spec screen, is there any commonality that exists between these sequences? Perhaps visualize the substrate sequences with a logo plot to see the prevalence of different amino acids around the target phospho-site and see if there is a consensus.

3. Summary of kinase substrate preference. I am a huge fan of the authors' Figure 1 & Supplementary Table, which summarize prior biosensors at the kinome scale. One suggestion for further improvement would be to also list the phosphopeptide sequences used in each biosensor and/or the consensus phosphorylation sites that are exploited in each case. This is not necessary, but it would be an incredible resource for students reading the paper.

Reviewer #2

(Remarks to the Author)

In this study, Bogomolovas and Chen develop a new genetically encoded fluorescent biosensor for protein kinase N (PKN) using an optimized substrate identification method. This method probes the entire cell contents for potential kinase substrates using phosphoproteomics to overcome the bottleneck of substrate identification. The specificity of this biosensor is demonstrated by assessing the biosensor response when AGC family kinases are activated. The PKN biosensor shows increased responses when PKN is overexpressed and shows decreased response when inhibited. The biosensor is then used to examine differences in plasma membrane and cytosolic PKN activity in HEK293 cells, revealing greater activity at the membrane.

I commend the authors on a very nice piece of work. The substrate mining method is clearly effective, given that the major hit from the screen yielded a functional and specific biosensor when the substrate was ported into the established ExRai backbone. However, the paper must be further strengthened by additional experiments, particularly stimulus experiments to activate PKN and demonstrate the full dynamic range of the biosensor and comparisons to phosphorylation, which is still a gold standard for activation. Some of the figures should also be refined to include more raw values, as the data presentation makes it difficult to fully assess the performance of the biosensor against negative controls.

Major points:

1. PKN Activation. For any newly developed biosensor, a thorough characterization of its dynamic range between fully inhibited and fully activated contexts is essential. The authors use inhibitor experiments to examine the “off state” of the biosensor, but only use PKN overexpression, not activation, to stimulate this kinase. However, protein kinases are typically highly regulated, so overexpression alone is unlikely to yield a state of maximum PKN activity. An ideal experiment would stimulate PKN activity to show the full dynamic range of the biosensor, for example by introducing a constitutively active RhoA. The response to such a PKN activator could also serve as a positive control for Figure 3A.

In addition, including a ground truth measure of PKN2 activity such as a Western blot or immunostaining for phospho-PKN2 in parallel to stimulation/overexpression experiments would more comprehensively show how the biosensor's dynamic range compares to this classical measurement of activity.

2. Non-normalized visualizations of biosensor activity. The authors show normalized data in time course experiments in which inhibitors are applied (Figure 3C, Figure 4A). This clearly illustrates that biosensor output falls with reduced PKN2 activity. However, it does not speak to the dynamic range of the biosensor. The R/R0 seems to be a normalized measurement to the value in the start frame of the time course, but not the baseline value (which R0 usually stands for). It is a bit unclear for readers if R/R0 == 1.00 is a state where PKN is fully activated or if it is baseline with low PKN activity. How does the baseline output of the wild-type PKN2 + biosensor compare to the non-catalytic PKN2KR case, or the non-phosphorylatable ExRai-PKNARTA biosensor? For example, in

figure 3C, it is nice and clear to see the control 'TA+PKN-WT' show no change upon drug inhibition, but it does not show if TA-mutant biosensor has a lower baseline output than a WT biosensor due to the normalization. To address that, it would be nice to show an unnormalized version of the plots, where WT-biosensor starts high and then goes down after drug inhibition, while the controls start low and keep low all along. It could be also useful to show the F/F0 measurements separately in 488 and 405 channels upon drug inhibition.

3. Motivation and discussion of PKN2. The authors describe the myriad roles that PKN plays in cellular function, however, this description did not feel compelling in motivating their choice to pursue creation of a PKN biosensor or their investigations into spatial differences in activity. It would greatly strengthen this paper if the authors further elaborated on the significance of PKN2 and their findings. Is there a canonical cell behavior that PKN2 is known to be involved in that motivates this work? Given the findings that PKN2 is active at the plasma membrane, is there evidence regarding what role it may be playing there? Is there a biological context in which PKN2 activity at the membrane is relevant?

4. Evidence on correct biosensor subcellular localization. In Figure 4B, normalized images are shown of the plasma-membrane-targeted biosensor. The findings regarding PKN activity differences in different subcellular locations should be substantiated by including raw images in either 488 or 405 channels to show that the PM-biosensor is actually localized on the membrane as a supplement to Figure 4B.

Minor points

1. Potential mislabelling in Figure 4A. The authors mention in text that one control is "inactive version of membrane targeted (PM-ExRai-PKNAR-TA)" (line 137-139). However the time course curve in Figure 4A might be mislabeled as "ExRai-PKNAR-TA". If this is not mislabeled, an PM-ExRai-PKNAR-TA is an appropriate negative control to include in this figure.

2. Demonstrate if the substrate consensus exists for PKN. There seems to be many hits showing strong phosphopeptide enrichment in WT-PKN pool from Figure 2B. The authors pick the top one for downstream biosensor development. I'm curious, for the top substrate hits that came out of your mass spec screen, is there any commonality that exists between these sequences? Perhaps visualize the substrate sequences with a logo plot to see the prevalence of different amino acids around the target phospho-site and see if there is a consensus.

3. Summary of kinase substrate preference. I am a huge fan of the authors' Figure 1 & Supplementary Table, which summarize prior biosensors at the kinome scale. One suggestion for further improvement would be to also list the phosphopeptide sequences used in each biosensor and/or the consensus phosphorylation sites that are exploited in each case. This is not necessary, but it would be an incredible resource for students reading the paper

Reviewer #3

(Remarks to the Author)

This manuscript performed an in vitro kinase reaction on cell lysate, detected phosphorylated peptides via mass spectrometry, selected one peptide to incorporate into a previously developed live cell ExRai biosensor of kinase activity, and characterized it within HEK293 cells.

I have several technical concerns about the work as follows:

Could the authors discuss or comment on how generalizable switching threonine for serine and the introduction of the +3 aspartate in the peptide are, i.e. what type of success rate might one expect for other peptide search and biosensor design campaigns? Did the authors try a certain number of peptides before finding one that works, and trying each of these changes? This would help readers understand how generalizable the approach is. Arguably, even if they tried one and it happened to work, this does not provide sufficient data to claim a generalizable method.

The authors wrote: "Maximum ratio changes were calculated as $\text{abs}(R_{\text{max}} - R_{\text{min}})/R_{\text{min}}$, where R_{max} and R_{min} are the average values pre- and post-inhibitor addition, respectively. The absolute value was used to handle occasional small negative values in non-responsive conditions." This use of absolute values seems inappropriate because it not only gets rid of the negative value but inverts the relative intensities of the R_{max} and R_{min} which could lead to very artifactual results. I would suggest reconsidering how the data are normalized, which is related to the next comment.

Fig 3A, is there validation of the introduction of ExRai-PKNAR into the cells? I.e. a positive control? Without a positive control, it is not clear if the negative signal is simply due to lower transfection efficiency of those plasmids.

Fig 3A and 3C: The normalization is a bit confusing. Wouldn't ExRai-PKNAR + PKN2-WT have a higher starting 488/405 ratio than the other two conditions? It seems everything is normalized within each sample to its own baseline, but panel 3B does not do this so is inconsistent. But also, this normalization hides important information about the actual activity level of the biosensor.

Fig 3A and 3C: The bounds around each trace were not defined in the figure caption.

Fig 3D: Why is the 488/405 ratio high for the top row (ExRai-PKNAR-TA) when the sensor peptide has been mutated and therefore should not bind the FHA1 domain?

Fig 3E: DeltaR I think should be negative?

Figure 4A. The inset immunofluorescence image of endogenous PKN2 localization is ambiguous. I.e. it is unclear what PKN2 signal is. If the signal is white, then it is hard to distinguish from background.

Figure 4B. Is there a reason the decrease in the ratio is substantially less than what was observed in Figure 3D?

The authors wrote: "In conclusion, these experiments demonstrate that ExRai-PKNAR is a specific and sensitive biosensor for PKN family kinases, capable of detecting changes in cellular activity of overexpressed PKN kinases." The data in Figure 3 did not probe specificity, only that it responded to PKN2 and inhibition of PKN2 by a chemical inhibitor. It did not assess responsiveness to other kinases.

Membrane targeting of ExRai-PKNAR could simply increase effective local concentration of the sensor with kinase leading to phosphorylation of the sensor. The fact that transient transfection of PKN could elicit signal throughout the cell, not just the membrane, suggests this is a concentration rather than specific membrane compartment effect. For example, if one were to concentrate the sensor at another organelle, would the same effect be seen? Given how much of the paper and discussion makes claims about the membrane compartment being the active compartment, this is important to know and prove.

Fig 4: Is the lack of response for the ExRai-PKNAR to the PKN2 chemical inhibitor consistent with the response observed in Figure 3?

In the discussion section, the connection made between ROCK and PKN2 seems a bit uncorrelated/unsubstantiated. It could help to provide more explanation of why this comparison was made.

"The constitutive activity of membrane-associated PKN2 we observed may thus be a key factor that explains its indispensability." This statement seems a bit tenuous to make even in the discussion section.

More details about the imaging methods need to be included, such as how background levels were assigned and subtracted/normalized out, whether these were done consistently across samples in some way. A range of exposure times was also listed in the methods section, how did this affect the comparability across samples and experiments? Was this done consistently?

"For membrane-targeted biosensors, a fine-tuned model was developed using a human-in-the-loop training approach from cyto2 model, specifically selecting cells with proper membrane localization of the biosensor." Is it clear if this human selection could have biased results towards those with positive biosensor signal at the membrane? I.e. should this membrane biosensor signal be also checked in the same single cells for subcellular spatial localization? This is related to questions about transfection efficiency and homogeneity.

I have concerns about the introduction, and about claims of the broad impact and originality of the work. I do not believe these are supported by the results presented. The technological advance/method is limited and similar to prior work using in vitro reactions performed on cell lysate. It also does not provide a comparison against prior methods to demonstrate an improvement in some metric of choice.

The claim of generalizability in the title and introduction is not supported by the work, which was limited in the number of peptides tested to one, in the controls used which did not demonstrate specificity, and in an understanding of what success rate is achieved by this method.

Font sizes in figures are quite small and required substantial zooming to read.

Did the substrates identified match that recently published by reference 40?

There are some non-grammatical typos throughout.

Version 1:

Reviewer comments:

Reviewer #1

(Remarks to the Author)

We commend the authors on a wonderful addition to the biosensor literature. All our concerns have been more than adequately addressed.

Reponses to reviewers' comments

Reviewer #1/#2 (Remarks to the Author):

In this study, Bogomolovas and Chen develop a new genetically encoded fluorescent biosensor for protein kinase N (PKN) using an optimized substrate identification method. This method probes the entire cell contents for potential kinase substrates using phosphoproteomics to overcome the bottleneck of substrate identification. The specificity of this biosensor is demonstrated by assessing the biosensor response when AGC family kinases are activated. The PKN biosensor shows increased responses when PKN is overexpressed and shows decreased response when inhibited. The biosensor is then used to examine differences in plasma membrane and cytosolic PKN activity in HEK293 cells, revealing greater activity at the membrane.

I commend the authors on a very nice piece of work. The substrate mining method is clearly effective, given that the major hit from the screen yielded a functional and specific biosensor when the substrate was ported into the established ExRai backbone. However, the paper must be further strengthened by additional experiments, particularly stimulus experiments to activate PKN and demonstrate the full dynamic range of the biosensor and comparisons to phosphorylation, which is still a gold standard for activation. Some of the figures should also be refined to include more raw values, as the data presentation makes it difficult to fully assess the performance of the biosensor against negative controls.

Major points:

1. PKN Activation. For any newly developed biosensor, a thorough characterization of its dynamic range between fully inhibited and fully activated contexts is essential. The authors use inhibitor experiments to examine the “off state” of the biosensor, but only use PKN overexpression, not activation, to stimulate this kinase. However, protein kinases are typically highly regulated, so overexpression alone is unlikely to yield a state of maximum PKN activity. An ideal experiment would stimulate PKN activity to show the full dynamic range of the biosensor, for example by introducing a constitutively active RhoA. The response to such a PKN activator could also serve as a positive control for Figure 3A.

We thank the reviewer for raising this important point regarding the characterization of the dynamic range of our biosensor. We agree that demonstrating both fully inhibited and fully activated states is essential for validating a new biosensor. In our study, we addressed the fully activated state by overexpressing the catalytic domain of PKN2, which we consider representing the maximally activated form of the kinase.

Our rationale is based on previous research demonstrating that truncation of PKN2 by removing the N-terminal 642 residues leads to a significant increase in its specific activity—approximately a 7-fold increase—as shown in Figure 3 of Matsubara et al. (PMID: 22511787). Furthermore, Lehtiö et al. (PMID: 9751706) reported that cleavage of PKN by activated caspase between residues 558-560, which liberates the C-terminal catalytic domain, results in more than a 30-fold increase in specific activity compared to the full-length protein (Figure 4 of that paper). These findings indicate that the isolated catalytic domain, as used in our study, functions as a constitutively active form of PKN.

This approach has been adopted in other studies as well. For example, the catalytic domain of PKN1 has been used as a constitutively active form in transgenic mice to investigate the in vivo consequences of PKN activity in different contexts (PMIDs: 17591691 and 20595653). Based on this body of evidence, we believe that overexpression of the catalytic domain of PKN2 effectively achieves maximal kinase activation in our experiments. We have incorporated this reasoning into the revised manuscript to provide additional clarity on our approach.

We hope this clarifies our rationale and demonstrates that our approach captures the fully activated state of PKN2, thereby enabling measurement of the biosensor's dynamic range.

In addition, including a ground truth measure of PKN2 activity such as a Western blot or immunostaining for phospho-PKN2 in parallel to stimulation/overexpression experiments would more comprehensively show how the biosensor's dynamic range compares to this classical measurement of activity.

We thank the reviewer for this excellent suggestion. To correlate our biosensor readout with PKN2 activity, we have chosen to perform an immunoprecipitation-in vitro phosphorylation assay. We opted for this method instead of assessing activation loop phosphorylation via Western blot because accumulating evidence—including our previous work—strongly indicates that, in the AGC family of protein kinases, activation loop phosphorylation is a prerequisite for kinase maturation rather than marker of activation (PMIDs: 33850054 and 11749377).

In our experiments, we immunoprecipitated wild-type (WT) and kinase-dead PKN2 from HEK293 cells. We then used these immunoprecipitates in an in vitro kinase assay with dephosphorylated, heat-inactivated HEK293T cell lysate serving as the substrate source, like the approach used in Figure 1 of our manuscript. This method has been successfully employed to measure PKN activity in other studies (PMIDs: 21037231 and 12783890).

As expected, the kinase-dead PKN2 showed no phosphorylation activity, while the WT PKN2 exhibited strong activity. These results correlate well with our biosensor readouts, thereby validating the biosensor's ability to accurately reflect PKN2 activity.

We have included these new data in the revised manuscript (see Figure 3G and corresponding text). We believe that this additional validation addresses the reviewer's concern and strengthens the evidence supporting our biosensor's dynamic range.

2. Non-normalized visualizations of biosensor activity. The authors show normalized data in time course experiments in which inhibitors are applied (Figure 3C, Figure 4A). This clearly illustrates that biosensor output falls with reduced PKN2 activity. However, it does not speak to the dynamic range of the biosensor. The R/R₀ seems to be a normalized measurement to the value in the start frame of the time course, but not the baseline value (which R₀ usually stands for). It is a bit unclear for readers if R/R₀ == 1.00 is a state where PKN is fully activated or if it is baseline with low PKN activity. How does the baseline output of the wild-type PKN2 + biosensor compare to the non-catalytic PKN2KR case, or the non-phosphorylatable ExRai-PKNARTA biosensor? For example, in figure 3C, it is nice and clear to see the control 'TA+PKN-WT' show no change upon drug inhibition, but it does not show if TA-mutant biosensor has a lower baseline output than a WT biosensor due to the normalization. To address that, it would be nice to show an unnormalized version of the plots, where WT-biosensor starts high and then goes down after drug inhibition, while the controls start low and keep low all along. It could be also useful to show the F/F₀ measurements separately in 488 and 405 channels upon drug inhibition.

We thank the reviewer for this excellent and insightful point regarding the visualization of our biosensor activity data. We understand that normalizing the data to the start frame (R/R₀) may obscure the baseline differences between the wild-type and mutant biosensors, making it unclear whether an R/R₀ value of 1.00 represents a state of full PKN2 activation or baseline activity with low PKN2 activity. This could also make it difficult to assess the dynamic range of the biosensor across different experimental conditions.

To address this concern, we have re-analyzed our data to include unnormalized fluorescence intensity measurements (new Figure 3C). This approach is consistent with methodologies used in previous studies employing ratiometric sensors (e.g., PMID: 30250062), and it allows for a more direct comparison of the biosensor activity without the potential confounding effects of normalization.

By presenting the unnormalized data, we are able to show that the wild-type PKN2 biosensor exhibits a higher baseline fluorescence ratio, which decreases upon drug inhibition. In contrast, the control conditions involving the non-catalytic PKN2^{KR} mutant and the non-phosphorylatable ExRai-PKN2^{R/A} biosensor maintain low fluorescence ratios throughout the time course. This clearly demonstrates that the TA-mutant biosensor has a lower baseline output than the wild-type biosensor, addressing the reviewer's point about baseline differences due to normalization.

We believe that these revisions satisfactorily address the reviewer's concerns and improve the overall transparency and interpretability of our data.

3. Motivation and discussion of PKN2. The authors describe the myriad roles that PKN plays in cellular function, however, this description did not feel compelling in motivating their choice to pursue creation of a PKN biosensor or their investigations into spatial differences in activity. It would greatly strengthen this paper if the authors further elaborated on the significance of PKN2 and their findings. Is there a canonical cell behavior that PKN2 is known to be involved in that motivates this work? Given the findings that PKN2 is active at the plasma membrane, is there evidence regarding what role it may be playing there? Is there a biological context in which PKN2 activity at the membrane is relevant?

Thank you for highlighting the need to better contextualize the biological significance of PKN2 and our findings. We have expanded our discussion to focus on two key aspects of PKN2 function where our biosensor could provide crucial mechanistic insights. PKN2 has been shown to orchestrate rear-end retraction during matrix-directed cell migration (Hetmanski et al., Dev Cell, 2019), but the spatiotemporal dynamics of its activity during this process remain unknown. Similarly, PKN2 plays a critical role in apical junction formation in epithelial cells (Wallace et al., Mol Cell Biol, 2011), but whether sustained PKN2 activity is required for junction maintenance or only initial assembly is unclear. Our finding of constitutive PKN2 activity at the plasma membrane provides a new framework for understanding these processes. We have revised the discussion to address these points and outline specific biological questions that can now be addressed using our biosensor.

4. Evidence on correct biosensor subcellular localization. In Figure 4B, normalized images are shown of the plasma-membrane-targeted biosensor. The findings regarding PKN activity differences in different subcellular locations should be substantiated by including raw images in either 488 or 405 channels to show that the PM-biosensor is actually localized on the membrane as a supplement to Figure 4B.

We thank reviewer for this suggestion. We have now added a representative raw fluorescence image from the 405 nm channel as an additional panel in Figure 4A, demonstrating proper plasma membrane localization of PM-ExRai-PKNAR.

Minor points

1. Potential mislabelling in Figure 4A. The authors mention in text that one control is “inactive version of membrane targeted (PM-ExRai-PKNAR-TA)” (line 137-139). However the time course curve in Figure 4A might be mislabeled as “ExRai-PKNAR-TA”. If this is not mislabeled, an PM-ExRai-PKNAR-TA is an appropriate negative control to include in this figure.

Thank you for spotting this labeling mistake, indeed “ExRai-PKNAR-TA” should read “PM-ExRai-PKNAR-TA”; we have corrected figure accordingly.

2. Demonstrate if the substrate consensus exists for PKN. There seems to be many hits showing strong phosphopeptide enrichment in WT-PKN pool from Figure 2B. The authors pick the top one for downstream biosensor development. I'm curious, for the top substrate hits that came out of your mass spec screen, is there any commonality that exists between these sequences? Perhaps visualize the

substrate sequences with a logo plot to see the prevalence of different amino acids around the target phospho-site and see if there is a consensus.

Thank you for your insightful comment regarding the potential existence of a substrate consensus sequence for PKN kinases. To investigate this, we generated sequence logos for two groups of peptides: a foreground set of 433 phosphopeptides that showed higher intensity in the WT-PKN-treated samples compared to the catalytically dead kinase, and a background set of 306 peptides with higher intensity in the catalytically dead kinase samples. Upon examining the sequence logos, we found that both the foreground and background sets displayed remarkably similar patterns when compared at bulk.

From this analysis we conclude that additional stratification metrics are needed to truly elucidate phosphorylation consensus motif from phosphoproteomics data. As a result, we concluded that substrates must be analyzed on an individual peptide basis. Therefore, we selected the top substrate hit from our mass spectrometry screen for downstream biosensor development.

3. Summary of kinase substrate preference. I am a huge fan of the authors' Figure 1 & Supplementary Table, which summarize prior biosensors at the kinome scale. One suggestion for further improvement would be to also list the phospho-peptide sequences used in each biosensor and/or the consensus phosphorylation sites that are exploited in each case. This is not necessary, but it would be an incredible resource for students reading the paper.

We thank reviewer for a great suggestion, we have included sequences with phosphosite marked for all reported kinase biosensors.

Reviewer #3 (Remarks to the Author):

This manuscript performed an *in vitro* kinase reaction on cell lysate, detected phosphorylated peptides via mass spectrometry, selected one peptide to incorporate into a previously developed live cell ExRai biosensor of kinase activity, and characterized it within HEK293 cells.

I have several technical concerns about the work as follows:

Could the authors discuss or comment on how generalizable switching threonine for serine and the introduction of the +3 aspartate in the peptide are, i.e. what type of success rate might one expect for other peptide search and biosensor design campaigns? Did the authors try a certain number of peptides before finding one that works, and trying each of these changes? This would help readers understand how generalizable the approach is. Arguably, even if they tried one and it happened to work, this does not provide sufficient data to claim a generalizable method.

The use of threonine in place of serine and the introduction of a +3 aspartate are well-established requirements for FHA domain recognition, which forms the foundation of numerous successful biosensor designs. These modifications are not unique to our work but rather represent a standard approach used in various FHA domain-based biosensors, including those for PKA, PKC, JNK, ERK, RSK, S6K, AMPK, and Akt [citations 12,16-19]. These previous studies have demonstrated that FHA domains consistently recognize phosphothreonine in the context of a +3 aspartate with high specificity. Therefore, these modifications represent a proven design principle rather than an empirical optimization specific to our sensor. Regarding the broader generalizability of our method, we refer to our new comparative analysis in Figure 5 and rebuttal below, which demonstrates that our approach successfully identified a peptide that outperforms both computationally predicted and literature-derived alternatives in this biosensor context.

The authors wrote: “Maximum ratio changes were calculated as $\text{abs}(R_{\text{max}} - R_{\text{min}})/R_{\text{min}}$, where R_{max} and R_{min} are the average values pre- and post-inhibitor addition, respectively. The absolute value was used to handle occasional small negative values in non-responsive conditions.” This use of absolute values seems inappropriate because it not only gets rid of the negative value but inverts the relative intensities of the R_{max} and R_{min} which could lead to very artifactual results. I would suggest reconsidering how the data are normalized, which is related to the next comment.

*Thank you for highlighting this important point regarding our data normalization method. We agree that taking the absolute value in calculating maximum ratio changes could potentially invert the relative intensities and introduce artifacts. To address this concern, we have reanalyzed the dataset presented in **Figure 4** without applying the absolute value to the maximum ratio change calculation. Specifically, we recalculated the maximum ratio changes as $(R_{\text{max}} - R_{\text{min}})/R_{\text{min}}$, preserving the sign of the changes. We compared the distributions of maximum ratio changes with and without the absolute value transformation. As shown in the figure below, the overall distribution remains similar (blue-original, red-after absolute value transformation) with minimal differences between the two methods and key distribution metrics (mean, median, standard deviation and IQR) are consistent between the two calculations. This suggests that the absolute value transformation did not significantly alter the data interpretation in our case. Furthermore, we plotted 100 individual traces from cells with negative Maximum ratio changes values and 100 traces with positive Maximum ratio changes values. The traces with negative maximum ratio changes predominantly originate from negative control groups without biosensor response, reflecting baseline noise. In contrast, the positive maximum ratio changes traces correspond to experimental groups showing a clear biosensor response upon inhibitor addition. To ensure robustness, we performed a sensitivity analysis using generalized linear mixed models with distributions capable of handling negative values, such as Gaussian and inverse Gaussian distributions. The results were consistent with those obtained using the Gamma distribution, confirming that our findings are not an artifact of the data normalization.*

Therefore, we have decided to retain our original analysis approach, including the use of the absolute value in calculating maximum ratio changes. We believe this method effectively handles small negative fluctuations due to noise in non-responsive conditions without inverting or misrepresenting the relative intensities in our data. This approach ensures consistent and accurate detection of biosensor responses across all experimental conditions.

PM-ExRai-PKNAR

ExRai-PKNAR

PM-ExRai-PKNAR^{TA}

Summary Statistics of Raw and Absolute new_deltaR by Subfolder

Subfolder	Sample Size (n)	Raw new_deltaR				Absolute new_deltaR			
		Mean (Raw)	SD (Raw)	Median (Raw)	IQR (Raw)	Mean (Absolute)	SD (Absolute)	Median (Absolute)	IQR (Absolute)
ExRai-PKNAR	1247	0.010	0.015	0.009	0.014	0.013	0.012	0.011	0.013
PM-ExRai-PKNAR	549	0.090	0.040	0.085	0.048	0.090	0.040	0.085	0.048
PM-ExRai-PKNAR TA	445	0.014	0.022	0.012	0.023	0.020	0.017	0.015	0.020

Fig 3A, is there validation of the introduction of ExRai-PKNAR into the cells? I.e. a positive control? Without a positive control, it is not clear if the negative signal is simply due to lower transfection efficiency of those plasmids.

Thank you for your valuable comment regarding the validation of ExRai-PKNAR introduction into the cells and the potential impact of transfection efficiency on the negative signal observed in Figure 3A. To address this concern, we performed an additional analysis using the data from Figure 3. Specifically, we generated a scatter plot of the emission at 405 nm (serving as a marker for reporter expression levels) versus the emission ratio of 488/405 nm (the reporter readout) at timepoint 0. Our results showed no significant correlation between the emission at 405 nm and the emission ratio of 488/405 nm, indicating that variations in reporter expression levels did not affect the sensor readout. This finding is consistent with previous reports on other ratiometric reporters [PMID: 11248055, 19122669, 21903779], which demonstrate that ratiometric measurements effectively mitigate artifacts caused by differences in indicator concentration. Therefore, we conclude that our ratiometric sensor provides reliable readouts independent of expression levels, supporting its validity in our experiments.

Fig 3A and 3C: The normalization is a bit confusing. Wouldn't ExRai-PKNAR + PKN2-WT have a higher starting 488/405 ratio than the other two conditions? It seems everything is normalized within each sample to its own baseline, but panel 3B does not do this so is inconsistent. But also, this normalization hides important information about the actual activity level of the biosensor.

Thank you for highlighting the confusion regarding the normalization in Figures 3A, 3B, and 3C. We acknowledge that using different normalization approaches between these panels may seem inconsistent and could potentially obscure important information about the biosensor's activity levels. In Figures 3A and 3C, which present time-course data showing the dynamic response of the biosensor over time, we normalized the fluorescence ratio ($R=F488/F405$) to its initial value at time zero (R_0), resulting in the normalized ratio R/R_0 . This normalization is a standard approach in the field of biosensor imaging (e.g., PMIDs: 32989297, 34963894, 35790710, 30250062), as it accounts for cell-to-cell variability and emphasizes the relative changes in biosensor activity over time.

In new Figure 3B, we provided the excitation ratio ($R = F488/F405$) without normalization to R_0 , since it represents static endpoint measurements. Because the biosensor response is inherently ratiometric, displaying the excitation ratio provides meaningful insights into the relative activity levels without additional normalization.

To address this concern, we have re-analyzed our data to include unnormalized fluorescence intensity measurements (new Figure 3C) for data presented in the original Figure 3C (revised Figure 3D). This approach is consistent with methodologies used in previous studies employing ratiometric sensors (e.g., PMID: 30250062), and it allows for a more direct comparison of the biosensor activity without the potential confounding effects of normalization. We are demonstrating that chemical inhibition of PKN2 in the WT PKN2 group brings biosensor readout down to levels of negative control, expressing catalytically dead PKN2, or inactive sensor.

Fig 3A and 3C: The bounds around each trace were not defined in the figure caption.

Thank you for pointing this out. The bounds around each trace in Figures 3A and 3C represent the 95% confidence intervals of the mean fluorescence ratio at each time point, calculated using the standard error of the mean. We've added this explanation to the figure captions to clarify.

Fig 3D: Why is the 488/405 ratio high for the top row (ExRai-PKNAR-TA) when the sensor peptide has been mutated and therefore should not bind the FHA1 domain?

Thank you for your insightful comment regarding Figure 3D. The high 488/405 ratio observed for the top row (ExRai-PKNAR-TA) represents the normalized fluorescence ratio R/R_0 , where R is the fluorescence ratio at each time point and R_0 is the initial ratio before any treatment. When there is no change in biosensor readout, the R/R_0 value remains around 1. In the case of the mutated reporter (ExRai-PKNAR-TA), the sensor peptide cannot be phosphorylated and bind FHA domain, so we expect no significant response to the PKN2 inhibitor. As a result, the R/R_0 ratio stays around 1, indicating no response of mutant biosensor. In contrast, with the non-mutated reporter, the addition of the inhibitor leads to a drop in the R/R_0 ratio, reflecting a decrease in biosensor readout due to PKN2 inhibition.

Fig 3E: DeltaR I think should be negative?

Thank you for your insightful comment regarding Figure 3E. You observed that ΔR should be negative based on our calculation method: ΔR was calculated as $abs(R_{max} - R_{min})/R_{min}$, where R_{max} and R_{min} are the average fluorescence ratios before and after inhibitor addition, respectively. We appreciate the opportunity to clarify this point. In our experiments, the inhibitor causes a decrease in the fluorescence ratio, so R_{min} is typically less

than R_{max} making $(R_{max} - R_{min})$ positive. Therefore, ΔR is inherently positive even without taking the absolute value. However, due to experimental noise or in non-responsive conditions, R_{min} may occasionally be slightly greater than R_{max} , resulting in a negative value for $(R_{max} - R_{min})/R_{min}$. To ensure that ΔR consistently reflects the magnitude of change regardless of direction, we take the absolute value of the calculation. This approach allows us to present all ΔR values as positive, indicating the extent of the biosensor's response to the inhibitor. Therefore, the ΔR values in Figure 3E are correctly displayed as positive. We have updated the Methods section to clarify this calculation and added an explanation in the figure caption to help readers understand why ΔR is presented as a positive value. Thank you again for your valuable feedback, which has helped us improve the clarity of our manuscript.

Figure 4A. The inset immunofluorescence image of endogenous PKN2 localization is ambiguous. I.e. it is unclear what PKN2 signal is. If the signal is white, then it is hard to distinguish from background.

Thank you for your comment. We have updated figure, to demonstrate PKN2 signal in non ambiguous way.

Figure 4B. Is there a reason the decrease in the ratio is substantially less than what was observed in Figure 3D?

Thank you for raising this important point. The smaller magnitude of change in Figure 4B compared to Figure 3D reflects the difference between measuring endogenous versus overexpressed PKN2 activity. In Figure 4B, we measure activity of endogenous PKN2, which is present at lower levels and not fully activated. In contrast, Figure 3D shows cells overexpressing just the catalytic domain of PKN2, resulting in both higher abundance and maximal activation of the kinase.

The authors wrote: “In conclusion, these experiments demonstrate that ExRai-PKNAR is a specific and sensitive biosensor for PKN family kinases, capable of detecting changes in cellular activity of overexpressed PKN kinases.” The data in Figure 3 did not probe specificity, only that it responded to PKN2 and inhibition of PKN2 by a chemical inhibitor. It did not assess responsiveness to other kinases.

Thank you for your valuable comment regarding the specificity assessment of ExRai-PKNAR. We recognize that establishing the biosensor's specificity is essential. In our study, we conducted several experiments to address this: Firstly, the peptide incorporated into ExRai-PKNAR was identified through assays specifically targeting PKN2, ensuring that it is a substrate for this kinase. We then tested the biosensor's responsiveness to activation by PKC using phorbol 12,13-dibutyrate (PDBu), since PKCs share significant catalytic domain similarity with PKNs. ExRai-PKNAR did not respond to PKC activation, suggesting minimal cross-reactivity with kinases closely related to PKNs. Additionally, we demonstrated that ExRai-PKNAR responds to a specific PKN2 inhibitor in HEK293T cells that do not overexpress PKN2, indicating that the biosensor can detect changes in endogenous PKN2 activity. The specificity of this inhibitor was validated using the DiscoverX KINOMEScan® panel, which screened it against 468 human kinases. The results confirmed that the inhibitor selectively targets PKN2 with high specificity (PMID: PMID: 35104640). While we acknowledge that no single experiment can definitively prove specificity, the combination of these findings provides strong evidence that ExRai-PKNAR is selective for PKN family kinases. We have updated the manuscript discussing the strengths and limitation of our approach.

Membrane targeting of ExRai-PKNAR could simply increased effective local concentration of the sensor with kinase leading to phosphorylation of the sensor. The fact that transient transfection of PKN could elicit signal throughout the cell, not just the membrane, suggests this is a concentration rather than specific membrane compartment effect. For example, if one were to concentrate the sensor at another organelle, would the same effect be seen? Given how much of the paper and discussion makes claims about the membrane compartment being the active compartment, this is important to know and prove.

We appreciate the reviewer's thoughtful comment about potential concentration effects in our membrane-targeting experiments. The experiments in Figure 4 specifically examine endogenous PKN2 activity, with cells expressing only the biosensor constructs, which is fundamentally different from our validation experiments in Figure 3 where PKN2 was overexpressed. Targeted biosensors are widely used to study compartmentalization of kinase signaling, with the most rigorous control being an inactive biosensor with identical localization. Indeed, our inactive plasma membrane-targeted biosensor showed no response to PKN2 inhibition. These findings, together with our immunofluorescence data demonstrating distinct membrane-associated and cytosolic pools of endogenous PKN2, strongly support authentic compartment-specific PKN2 activity at the plasma membrane rather than a concentration-dependent artifact.

Fig 4: Is the lack of response for the ExRai-PKNAR to the PKN2 chemical inhibitor consistent with the response observed in Figure 3?

We thank the reviewer for this important question about the apparent differences in ExRai-PKNAR responses between Figures 3 and 4. The different responses of ExRai-PKNAR to PKN2 inhibitor reflect two distinct experimental contexts. In Figure 3, we overexpressed the isolated catalytic domain of PKN2, which lacks the regulatory domains that mediate membrane binding and regulate kinase activity (PMID: 15375078). This overexpressed catalytic domain is constitutively active and distributed throughout the cytoplasm, leading to robust sensor phosphorylation regardless of cellular location. In contrast, Figure 4 examines the activity of endogenous PKN2, which contains all regulatory domains and exists in two distinct pools as shown by our immunofluorescence data - membrane-associated and cytoplasmic. Our findings that only the membrane-targeted sensor detects constitutive PKN2 activity align with previous work on PKN1 (PMID: 15375078), where biochemical approaches demonstrated that the membrane-associated pool of endogenous PKN1 is specifically active compared to its cytoplasmic counterpart. Thus, the lack of cytoplasmic ExRai-PKNAR response to PKN2 inhibitor in Figure 4 is consistent with compartment-specific regulation of endogenous PKN2 activity, in contrast to the ubiquitous activity of overexpressed catalytic domain seen in Figure 3. We have added this clarification in the Discussion section to help readers better understand these distinct experimental contexts and their implications.

In the discussion section, the connection made between ROCK and PKN2 seems a bit uncorrelated/unsubstantiated. It could help to provide more explanation of why this comparison was made.

We thank the reviewer for this comment about the ROCK2-PKN2 comparison. We agree that this connection deserves more explicit explanation. The parallel we draw is based on two key observations: First, both PKN2 and ROCK2 display constitutive catalytic activity at the membrane without requiring specific activating signals, as demonstrated for ROCK2 [PMID: 26620183]. This observation is further supported by the strong effects of inhibitors compared to potential activators in previous ROCK biosensor work [PMID: 27885213, doi: <https://doi.org/10.1101/2024.05.30.596680>]. This is distinct from many other kinases like PKA or PKC that require specific activators to elicit biosensor responses. While most biosensor studies focus on kinase activation, our work, similar to ROCK studies, highlights the importance of considering basal kinase activity and its spatial organization. This approach of using inhibitors rather than activators to reveal constitutive activity may be valuable for understanding compartment-specific regulation of other basally active kinases. We have expanded the discussion to better articulate these mechanistic parallels and broader implications for studying kinase regulation.

The constitutive activity of membrane-associated PKN2 we observed may thus be a key factor that explains its indispensability.” This statement seems a bit tenuous to make even in the discussion section.

We thank the reviewer for this observation. You are right - the statement about PKN2's constitutive activity and indispensability should be more measured. We have adjusted our discussion to simply note that the relationship

between PKN2's constitutive membrane activity and its essential role in development warrants further investigation.

More details about the imaging methods need to be included, such as how background levels were assigned and subtracted/normalized out, whether these were done consistently across samples in some way. A range of exposure times was also listed in the methods section, how did this affect the comparability across samples and experiments? Was this done consistently?

Thank you for raising these important methodological points. We have expanded our methods section to provide more detailed information about image processing and analysis. Our approach ensures consistent analysis of only expressing cells by implementing thresholding to exclude background pixels (set to NaN values). To validate our methodology, we performed sensitivity analysis using different threshold levels (50, 100, and 200 photons per pixel) on images obtained using Leica SP8 system. The ratiometric measurements showed consistent results across these thresholds, which is expected since ratiometric reporters effectively mitigate artifacts caused by differences in indicator concentration or expression levels. This robustness to expression level variations is a well-established advantage of ratiometric measurements. We have now expanded our methods section to include these details and better describe our image processing workflow.

For membrane-targeted biosensors, a fine-tuned model was developed using a human-in-the-loop training approach from cyto2 model, specifically selecting cells with proper membrane localization of the biosensor.” Is it clear if this human selection could have biased results towards those with positive biosensor signal at the membrane? I.e. should this membrane biosensor signal be also checked in the same single cells for subcellular spatial localization? This is related to questions about transfection efficiency and homogeneity.

Thank you for this important point about potential selection bias. We want to clarify that our human-in-the-loop training for the Cellpose model was focused solely on proper membrane targeting (i.e., cells showing clear membrane localization of the biosensor based on the fluorescence signal distribution), not the biosensor activity (ratiometric) signal. The selection criteria were applied to the raw fluorescence images in a single channel before any ratio calculations were performed, making it impossible to bias selection based on biosensor activity. This approach is similar to standard practice in the field where proper subcellular targeting of biosensors must be verified.

I have concerns about the introduction, and about claims of the broad impact and originality of the work. I do not believe these are supported by the results presented. The technological advance/method is limited and similar to prior work using in vitro reactions performed on cell lysate. It also does not provide a comparison against prior methods to demonstrate an improvement in some metric of choice. The claim of generalizability in the title and introduction is not supported by the work, which was limited in the number of peptides tested to one, in the controls used which did not demonstrate specificity, and in an understanding of what success rate is achieved by this method.

Thank you for these suggestions to validate our method's utility. We have addressed these concerns by adding a new Figure 5 that directly compares three different approaches for PKN substrate peptide selection. We evaluated (1) our modified KESTREL-derived peptide (PKNAR), (2) a computationally predicted optimal peptide based on published algorithm/dataset [PMID: 36631611], and (3) a previously validated PKN substrate peptide from literature [PMID: 21749319]. The biosensor variants were constructed by replacing the substrate sequence in ExRai-PKNAR while maintaining all other components identical. Despite the computational approach yielding a peptide with the highest predicted specificity scores and the literature peptide being previously validated biochemically, our PKNAR peptide demonstrated superior performance in the context of a live-cell biosensor. Upon PKN2 inhibitor treatment, PKNAR showed the most robust response, while the computationally predicted and literature-derived peptides exhibited significantly smaller changes in biosensor signal. This comparative analysis empirically validates our peptide identification strategy and demonstrates its

effectiveness in identifying sequences specifically suited for biosensor applications. These results have been particularly revealing as they show that computational predictions or biochemical validation alone may not fully predict a peptide's performance in the complex cellular environment of a conformational biosensor. We have revised the manuscript to include these comparative data and provide a more nuanced discussion of our method's practical utility in biosensor development.

Font sizes in figures are quite small and required substantial zooming to read.

We thank reviewer for raising this issue. We have increased font size in our figures

Did the substrates identified match that recently published by reference 40?

We thank reviewer for the excellent question. We believe that our new Figure 5 and accompanying text addresses this issue.

There are some non-grammatical typos throughout.

We thank the reviewer for this comment. We have thoroughly proofread our revised manuscript and corrected the typos.

Response to Reviewers' Comments

Reviewer #1: *We commend the authors on a wonderful addition to the biosensor literature. All our concerns have been more than adequately addressed.*

We sincerely thank the reviewer for their positive assessment of our work and their kind words about our contribution to the biosensor literature. We are grateful for all the constructive feedback provided throughout the review process, which has helped us improve the quality of our manuscript.